# Role of Precision Oncology in Type II Endometrial and Prostate Cancers in the African Population: Global Cancer Genomics Disparities

**DOI:** 10.3390/ijms23020628

**Published:** 2022-01-06

**Authors:** Rahaba Marima, Rodney Hull, Mandisa Mbeje, Thulo Molefi, Kgomotso Mathabe, Abdulrahman M. Elbagory, Demetra Demetriou, Zodwa Dlamini

**Affiliations:** 1SAMRC/UP Precision Oncology Research Unit, Pan African Cancer Research Institute (PACRI), University of Pretoria, Hatfield 0028, South Africa; rahaba.marima@up.ac.za (R.M.); rodmey.hull@up.ac.za (R.H.); mandisambeje@gmail.com (M.M.); thulo.molefi@up.ac.za (T.M.); kgomotso.mathabe@up.ac.za (K.M.); amn.elbagory@tuks.co.za (A.M.E.); demetrioudemetra223@gmail.com (D.D.); 2Department of Medical Oncology, University of Pretoria, Hatfield 0028, South Africa; 3Department of Urology, Faculty of Health Sciences, University of Pretoria, Hatfield 0028, South Africa

**Keywords:** precision medicine, precision oncology, pharmacogenomics, type II endometrial cancer (ECa), prostate cancer (PCa), low middle income countries (LMICs), African population

## Abstract

Precision oncology can be defined as molecular profiling of tumors to identify targetable alterations. Emerging research reports the high mortality rates associated with type II endometrial cancer in black women and with prostate cancer in men of African ancestry. The lack of adequate genetic reference information from the African genome is one of the major obstacles in exploring the benefits of precision oncology in the African context. Whilst external factors such as the geography, environment, health-care access and socio-economic status may contribute greatly towards the disparities observed in type II endometrial and prostate cancers in black populations compared to Caucasians, the contribution of African ancestry to the contribution of genetics to the etiology of these cancers cannot be ignored. Non-coding RNAs (ncRNAs) continue to emerge as important regulators of gene expression and the key molecular pathways involved in tumorigenesis. Particular attention is focused on activated/repressed genes and associated pathways, while the redundant pathways (pathways that have the same outcome or activate the same downstream effectors) are often ignored. However, comprehensive evidence to understand the relationship between type II endometrial cancer, prostate cancer and African ancestry remains poorly understood. The sub-Saharan African (SSA) region has both the highest incidence and mortality of both type II endometrial and prostate cancers. Understanding how the entire transcriptomic landscape of these two reproductive cancers is regulated by ncRNAs in an African cohort may help elucidate the relationship between race and pathological disparities of these two diseases. This review focuses on global disparities in medicine, PCa and ECa. The role of precision oncology in PCa and ECa in the African population will also be discussed.

## 1. Introduction

Sir William Osler (1849–1919) famously stated that no two persons can react alike to disease [1]. Precision oncology is derived from precision medicine, a form of medicine where information about an individuals’ genetic make-up is used to personalise or tailor medical care specifically for the individual. This can include the use of protein or RNA biomarkers for disease prevention, early diagnosis, improved prognosis and treatment. For this reason, precision medicine is also known as personalized medicine. The application of precision medicine in cancer has proved to be a popular research goal and led to the adoption of the term precision oncology [2,3]. The customization of therapeutic regimens to each patient is important because of the unpredictable effects induced by therapeutic drugs that can vary from one person to another according to their own genetics, phenotypic, or psychosocial characteristics [4]. For example, the HIV drug abacavir and the antiseizure drug carbamazepine should not be prescribed to patients who carry the HLA-B*5701 or HLA-B*1502 genotype to avoid serious side effects [5]. As such, precision medicine enables health care workers to make data-driven decisions. For example, specific target genes that increase a patient’s risk of developing a certain cancer can assist in early diagnosis, while those genotypes that predict better or worse outcomes can improve prognosis and, finally, transcriptomic changes can predict the therapeutic effects of drugs for each patient [5]. Pharmacogenomics is the cornerstone of precision medicine and is a research field concerned with how a person’s unique genome affects their response to drugs [6]. It has been reported that, after heart failure and cancer, medical error is the third leading cause of mortality in the USA., with 180,000 to 251,000 medical error-related deaths annually [7,8]. Factors contributing to this devastating number include misdiagnosis, medical system communication breakdown, growing cost and poorly coordinated care. In combating these devastating effects, precision medicine is evolving as a promising innovative pillar for transforming health care and improving overall patient outcome [9,10]. Precision medicine integrates multi-omics profiles with demographic and epidemiological data as well as clinical and imaging data. Not only does this multifaceted integration allow for the aforementioned improvement in early diagnosis, prognosis and treatment, but it also allows for lower cost treatment and disease management. Furthermore, the efficiency of this integration primarily depends on the 4Ps: Predictive, Preventative, Personalised and Participatory treatment for each individual patient [11]. Advances in precision medicine are being actively pursued, despite various challenges such as ethical and social issues and the protection and privacy of patients’ omics data [12].

Precision medicine relies on how an individual’s unique molecular profile influences the individual’s susceptibility to disease and response to medical treatment [13]. Over the past few years, there has been a paradigm shift from a ‘one drug fits all’ traditional approach to a personalized patient orientated approach [14]. It has been reported that about 95% of an individuals’ drug response is attributed to genetic elements [15]. Furthermore, genetic factors have also been reported to contribute to about 20% of the total reported cases of adverse drug reactions (ADR) [16,17].

The World Bank defines low middle income countries as those where the Gross National income (GNI) is between $1036 and $4045 per capita, while the High-income countries are defined as those with a GNI between $4046 and $12,535. These labels are those set by the World Bank in 2021 and are regularly re-assessed based on the changing economy of the world [18]. Prostate cancer (PCa) is one of the leading causes of deaths in men globally and in the African population. High income countries (HICs) have higher PCa incidence rates than low middle income countries (LMICs). Despite this, HICs have lower PCa mortality rates compared to LMICs. The highest mortality rates for PCa are found in sub-Saharan Africa (SSA) and emerging research reports that a link exists between PCa and African ancestry [19]. While data are currently available on PCa in African men, there is still a lot we do not know about the role of the patient’s genome in the development and progression of PCa in African men [20]. Since African men have a poor PCa prognosis compared to their Caucasian counterparts, it is suspected that genetic differences resulting from different ancestry may play an important role in this difference (reviewed in [2]). PCa is not the only sex specific cancer where African populations have higher incidence and mortality rates. Type II endometrial cancer (ECa) in African women has a similar pattern, with a poorer prognosis and worse outcome compared to their Caucasian counterparts [21]. Once again, there is an absence of genomic data from Africans to adequately link type II ECa to African ancestry [22]. The data we currently have show that there are unique gene profiles in African women that are not seen in Caucasian women. It is hoped that these differences may serve as targets for unique molecular therapies [23]. It is well known that populations described as African have the highest levels of genetic variations within them [24]. Despite this potential wealth of genetic data present in African populations, due to the high genetic diversity of African populations, and LMICs with African populations, such groups/populations are still underrepresented in the international platforms of genome-associated research studies [25]. This review will discuss the global disparities in the effective diagnosis and treatment of PCa and ECa. These two cancers have been shown to be linked to genetic differences in African populations [26,27] and as such are prime targets for the application of precision oncology. For this reason, this review will also discuss the use of precision oncology as a tool to fight these two sex specific cancers, both of which seem to be more devastating to those of an African ancestry due to their specific genomic profiles.

## 2. GLOBOCAN 2020 Stats for PCa and ECa

An analysis of the incidence rates of PCa in the GLOBOCAN database [28] shows the stark contrast between HICs and LMICs, with “western” HICs such as Australia, the United Kingdom, and the United States of America, having very high incidence rates of PCa. Non-western HICs such as Poland and some non-Western upper middle income countries such as Russia have lower incidence rates. Countries such as Jamaica have extremely high incidence rates despite being an LMIC [28] (Figure 1A,C). A similar situation is observed in other similar Caribbean countries such as Saint Lucia and the Dominican Republic (both are upper middle income countries). Among the LMIC regions, the southern tip of Africa has higher than average incidence rates with South Africa, Namibia (both upper middle income countries), Zambia, and Zimbabwe (both lower middle income countries) having higher incidence rates than most of the surrounding countries [28]. These differences may be due to socio-economic, cultural, or genetic factors. It is interesting to note that Caribbean countries such as Cuba have a far lower incidence rate than some of the surrounding island nations, which may point to a genetic component involving African ancestry [29]. The mortality rates show that, despite the high incidence in HICs, they have a lower mortality rate than that seen in LMICs [28]. For instance, despite having a lower incidence rate than that observed in the USA, South Africa has nearly twice the mortality rate (Figure 1B,C). The mortality rates shown in Figure 1B show that the highest mortality rates generally occur in central and sub-Saharan Africa—with some other countries outside this region such as Venezuela and Indonesia also having high mortality rates. The variations observed between different LMICs also highlights the importance of variations within large population groups such as those commonly defined as African. [24]. The high level of genetic variation in populations described as Africa may help to explain the differences observed in the relative incidence and mortality rates observed in South Africa, Nigeria and Kenya. Both Nigeria and Kenya are defined as low middle income countries, while South Africa is defined as an upper middle income country. Despite having a much higher incidence rate than either Kenya or Nigeria, South Africa has comparable mortality rates to both these countries. The fact that African populations are more susceptible to increased mortality due to prostate cancer can be seen in the case of Brazil. This upper LMIC has an incredibly high incidence rate but a lower mortality rate than any of the African LMICs represented here [28].

GLOBOCAN includes endometrial cancers under the umbrella term Corpus Uterine cancers [28]. Endometrial cancers are the most common gynecological cancers in HICs. This can be seen in the data presented in Figure 2, where the highest incidence rate is observed in the HICs United States of America, the United Kingdom, Australia and the upper LMIC Russia [28]. The LMICs South Africa, Namibia and Zimbabwe have some of the highest incidence rates in Africa, exceeded only by Benin and Mauritius. However, the HICs with high incidence rates still have higher mortality rates than these LMICs [28] (Figure 2B). Despite this, the relative mortality is still very high in these LMICs. For instance, despite the USA having nearly two and a half times the incidence of endometrial cancer than South Africa, the mortality rate is only one and a half times that of South Africa’s (Figure 2C). The mortality rates in both LMICs Jamaica and Samoa are exceedingly high [28].

## 3. Prostate Cancer (PCa)

### 3.1. Diagnosis and Risk Factors

Prostate cancer is often only diagnosed at a very late stage because the initial stages of the disease have no symptoms [30]. If symptoms do occur, they include dull pain in the lower abdomen; frequent urination; pain during urinating; blood in the urine; pain during ejaculation, loss of weight and appetite and bone pain. The most common screening tests for prostate cancer are the digital rectal exam (DRE) and prostate specific antigen (PSA) blood test [31]. The DRE is a physical examination of the prostate, where the size and shape or thickness of the prostate can give an indication of prostate cancer. While the test is easily performed and cost effective, it may not be able to detect early-stage prostate cancer [32]. PSA is produced by the prostate and is over-produced by prostate cancers. The PSA blood test measures the level of PSA in the blood, with high levels indicating possible prostate cancer. Despite the test being easy to perform and relatively cheap, it does not provide any information of the type of cancer and cannot distinguish between prostate cancer, benign enlargement of the prostate or inflammation of the prostate [32,33].

Once these initial studies have indicated the presence of PCa, further diagnostic tests can be carried out and these include transrectal ultrasound, magnetic resonance imaging (MRI) or prostate biopsy. These techniques require specialised equipment and trained practitioners and may be limited in LMICs [34]. The aggressiveness of the PCa is commonly determined through genomic testing, where the presence of specific genetic mutations can indicate the aggressiveness of the cancer [31]. However, many of the specific genes these tests look for are based on research performed on non-African individuals, and those of African ancestry may have different genetic mutations that can drive the aggressiveness of the cancer [32,33].

Environmental factors such as diet have been implicated in contributing to PCa incidence. Men who emigrate from a country with a low incidence of PCa tend to develop PCa at the same rate as men in their adopted country. This implicates the change in environment between the native low incidence country to that of the high incidence adopted country as a risk factor [35]. One of the most obvious changes in the environment would be the change in diet. The Western diet is associated with higher PCa risk and is high in fat, red meat, alcohol, and dairy products [36]. High meat intake is suspected to play a role in more aggressive PCa while increased fruit and whole grain food is associated with decreased PCa risk [37]. Obesity is suspected to contribute to the progression of PCa as there is a correlation between PCa progression and body mass index [35,36,37,38]. It is thought that this progression of PCa and increased incidence of PCa linked with obesity is due to the hormone changes induced by excess fat deposits [38].

### 3.2. Genomics, Racial and Socioeconomic Disparities

Therapeutic clinical trials for men with PCa have considerably increased over the past 16 years [39]. However, one of the obstacles impeding significant progress is the lack of adequate representation of other populations such as those of African ancestry in general medical research (Figure 3A). In a recent paper describing the methods of representing genome wide association study (GWAS) data, a summary of the race of the participants in these studies showed the racial disparities in these recent GWAS. This showed an obvious bias towards white or European populations in these studies. This is represented in Figure 3A. In addition to this, the enrolment of participants in the clinical trials for three FDA approved drugs specifically for PCa treatment is shown in (Figure 3B). The data in Figure 3B show the percentage of each racial group enrolled in these clinical trials. For instance, PCa-centered studies in the USA have revealed the unwillingness of African American men to partake in such clinical research given the opportunity. Concerns over transparency and relevance to cultural contexts need to be considered for adequate inclusion [40].

The United Nations (UN) has developed the Human Development Index (HDI) as a statistic to measure a country’s level of social and economic development. The HDI is made up of various measurements, The mean years of education, life expectancy and gross national income per capita [41]. A study performed by Sharma in 2019 to identify if there is a relationship between HDI and the burden of prostate cancer, in the form of mortality-to-incidence ratio, used pairwise correlation and bivariate regression [30]. This was performed in 87 countries, in the period 1990–2016. It was found that countries with a lower HDI had higher mortality and lower survival rates. However, the mortality rate was shown to decrease over the period of the study. This is probably due to advancements in screening and treatment which have become available even in low income countries [30].

In precision oncology, the best interpretation of genomic results is achieved through multidisciplinary input. This also reduces bias and uncertainty in the clinical data [3]. Many genetic studies have identified PCa being linked to specific loci, Table 1. The majority of these studies were once again initially performed in men of European or Asian descent and many of these described loci have not been identified as being linked to PCa in men of African descent (PCa-African loci) [42,43,44,45,46,47]. Some of the loci associated with PCa in men of European descent showed much lower effects, no effects, or even completely opposite effects in men of African ancestry [48]. However, many of the PCa associated loci have been identified as being shared between men with PCa of both European and African ancestry. These include 8q24, 3p12 [43], *KLK2/3* (19q13.33), *NUDT10/11* (Xp11.22) [42], 11q13.2, *HNF1B/TCF2* (17q12) [49], *JAZF1*, and *MSMB* [50]. Table 1 shows the PCa population linked genetic loci.

Previous studies involving varied populations, but still mainly European men, have identified genes that are associated with PCa include those involved in DNA damage and repair, carcinogen metabolism, inflammation and steroid hormone metabolism [60,61,62,63,64,65]. Specific genes with mutations and expression changes associated with PCa include androgen receptor (*AR),* telomerase-related genes (*TERT*, *TET)* [66], genes involved in carcinogen metabolism such as UDP-glucuronosyltransferase 1–8 *(UGT1A8)* and cytochrome P45021A2 (*CYP21A2)*, metalloproteinase genes and various non-coding RNAS (ncRNAs), including micro-RNAs (miRNAs) and long non coding RNAs (lncRNAs) [67]. By studying and comparing the mutation profiles of 474 genes in different stage tumors from patients of European, African and Asian ancestry, Mahal et al. noted that men of African descent had higher rates of forkhead box protein A1 (*FOXA1)* mutations [68]. FOXA1 is a transcription factor that is responsible for tissue-specific gene expression and regulation of gene expression in differentiated tissues. Therefore, mutations that could potentially affect the function of this transcription factor in cancer is not surprising. This study also identified that African men with metastatic PCa had higher rates of mutations in androgen receptor genes as well as genes involved in DNA-repair [68].

A similar study comparing mutation profiles in men of African and European ancestry with PCa was conducted. This study focused on genes involved in immune-oncogenic pathways. A race specific gene expression profile was identified with 38 differentially expressed genes specific to each race group. These genes were involved in immune-oncogenomic pathways such as cytokine signaling, interferon (IFN) signaling (IFNγ and IFNα responses), apoptosis, nuclear factor NF-kB (NF-kB) signaling in the tumor necrosis factor alpha (TNFα) pathway, epithelial–mesenchymal transition (EMT) pathways, and signaling by ILs including IL4 and IL13. This implies that immune related pathway alteration is more prevalent in men of African ancestry with PCa [69].

Testosterone metabolism plays an important role in PCa and the cytochrome P450 enzymes involved in this metabolic pathway show high levels of allelic variations depending on ethnicity and ancestry [70,71]. The androgen receptor, a well- known molecular participant in PCa, was found to have different polymorphisms which occur at different frequencies in men with PCa depending on ethnicity and ancestry. These polymorphisms involve high-frequency repeats that occur in the amino-terminal. Men of African ancestry typically have shorter CAG repeats [72]. Race specific changes in the DNA methylation levels of certain PC were also identified in malignant PCa [73,74]. This leads to gene silencing, and higher rates of methylation were observed on the regions coding for CD44 and glutathione S-transferase P (GSTP1) in men of African ancestry with PCa. The silencing of these genes is associated with increased risk of PCa [74,75].

## 4. Endometrial Cancer (ECa)

### 4.1. Diagnosis and Risk Factors

The symptoms of endometrial cancer include post-menopausal bleeding, bleeding between periods, pelvic pain and abnormal discharge. It is advised that women with these symptoms should be screened, but there is no evidence that asymptomatic women should be screened [76]. High risk patients should be screened annually after the age of 35 [76]. Physical examination should initially be performed to eliminate any other causes of the symptoms. This can be followed by the most common and recommended diagnostic tests, transvaginal ultrasonography with endometrial biopsy. Transvaginal ultrasonography is commonly available and cheap and sensitive. It measures endometrial thickness and any measurement of a thickness higher than 5 mm is an indicator of endometrial pathology [77]. Endometrial tissue biopsy is the most reliable diagnostic test; however, it may not always be easy to obtain a sufficient sample. Options to increase tissue sampling include dilatation and curettage (D&C), through the use of a curette. Sampling or diagnosis can be achieved with a specialized sampling tool known as the Pipelle. This is accurate and cost effective [78]. Other diagnostic options include hysteroscopy and, although MRI, positron emission tomography (PET) and computed tomography (CT) scans are also useful diagnostic tools, they are expensive and may not be readily available in low income countries [77].

Even in higher income countries, such as the U.S.A., patient mortality and positive treatment outcome rely on the early diagnosis of endometrial cancer. For instance, a comparison between the mortality rates in rural and urban endometrial cancer patients in Utah showed that rural women had a higher mortality rate as a result of a later diagnosis of the disease. This is most likely due to screening facilities not being readily available in rural areas, requiring patients to take more time and effort, travelling a greater distance [79].

In African American communities, the best predictors of endometrial cancer related death (survival) were early stage diagnosis. This was followed by family income and the financial well-being of the family and body mass index (BMI). However, further analysis revealed that the only stage of the cancer and BMI are accurate predictors of patient survival [80]. BMI is related to socioeconomic status while obesity (high BMI) is associated with increased risk of developing cancer and increased mortality [81]. High BMI is also associated with increased mortality following surgical treatment of endometrial cancer [82].

### 4.2. Genomics, Racial and Socioeconomic Disparities

Histologically, ECa is classified into either type I, estrogen dependent with a better outcome or type II, estrogen-independent with a poor prognosis. However, recently, histopathological and molecular reports have indicated a rather complex ECa risk stratification approach [83]. Results from The Cancer Genome Atlas (TCGA) research network have established four distinctive molecular subtypes: ultra-mutated, hyper-mutated, copy-number low and copy-number high [23,83]. Each of these four sub-types reflect the underlying molecular alterations and associated clinical phenotypes. The recent molecular classification of ECa presents with better opportunities to understand ECa tumor biology, differentiated risk stratification, improved prognosis and improved estimation of the responses of a patient to therapy [23,83]. The ultimate goal of this new risk stratification is its integration into clinical settings and thereby create a foundation for precision oncology in ECa patient care, particularly type II ECa [23]. Type II ECa survival and disease have been shown to be correlated with household income, with women from households with a higher income presenting with less aggressive forms of the disease, disease at an earlier stage and increased survival [84]. One useful indicator of the socioeconomic status of an individual is their level of education. As such, it was also found that those women with higher levels of education are less likely to only be diagnosed at the later stages of the disease and are more likely to get effective treatment and consequently have higher rates of survival [85]. This socioeconomic effect is amplified in poor black women, who are more than twice as likely to die from type II ECa. Apart from any role played by genetics or family history, this is most likely due to the fact that Black women more often receive treatment at a much later stage of disease [86].

Although recent advances have identified a number of molecular targets, which are currently being explored for effective treatment of type II ECa, there is still a lack of novel biomarkers and therapeutic targets. P53 mutations have been reported in 57.7–92% of type II ECa [87].

Studies have identified multiple genes whose expression changes in type II ECa. Oncogenes whose expression changes include GTPase kras (*KRAS)*, human epidermal growth factor receptor 2 (*HER2)*, epithelial growth factor receptor (*EGFR*), phosphatidylinositol 3-kinase catalytic subunit (*PI3KCA*) and fibroblast growth factor receptor 2 (*FGFR2*). Additionally, the expression of tumor suppressors, such as PTEN, p53, p21 and cyclin-dependent kinase inhibitor 2A (CDKN2A), is also altered in type II ECa. Other genes whose expression is altered in type II ECa include genes involved in apoptosis, genes involved in DNA mismatch repair and genes coding for hormone receptors (*BCL 2, hMLH1, hMSH2, hMSH6, PMS1 and PMS2, ER and PR*)) [88,89].

The change in the expression of some genes is so marked that they can be used as markers for ECa; these include the proliferation marker Ki-67 and angiogenesis growth factors (VEGF-A) [90]. More ECa include genetic alterations in p53, HER2, p16 and E-cadherin [91]. The use of changes in p53 expression was particularly noticeable in African American ECa patients compared with those of European ancestry [91]. As recently discovered functional parts of the human genome, the non-coding RNAs (ncRNAs) including microRNAs and long-ncRNAs have been reported to play a role in tumor development, progression and drug resistance. Furthermore, the ncRNA signatures in distinct races in PCa and ECa remain to be elucidated [92,93].

## 5. Non-Coding RNAs in PCa and ECa

### 5.1. PCa and Type II ECa Associated Micro-RNAs (miRNAs) in African Population

MicroRNAs (miRNAs) act as regulators of gene expression by binding to specific mRNAs and inhibiting or modifying their translation. They are able to target specific mRNAs by binding to complementary regions in the 3’ untranslated region (UTR) of the target mRNAs [94] (Figure 4). Aberrant miRNA expression has been observed in PCa [95,96] and ECa [21]. In both cancers, these miRNAs can lead tumor progression or tumor suppression.

The PCa differences in the miRNA profiles of men of European ancestry and men of African ancestry have been identified that are related to the occurrence, prognosis and progression of PCa [97]. Five miRNAs were identified to have different expression in men of African and European ancestry. These were miR-1b, miR-26a, miR-30c-1, miR-219 and miR-301 [97]. MiRNA26a expression increased in African American non-malignant, malignant, and metastatic prostate cancer cells, compared to European non-malignant, malignant, and metastatic prostate cancer cells. The expression of the miRNA increased in the cell lines of both African and European origin as the malignancy of the cells increased [98,99]. Decreased miR-26a levels lead to cell cycle arrest at G2/M phase followed by caspase 3/7 activation [100]. PCA patients of African ancestry and European ancestry have also exhibited differential expression of let7c and miR-30c [101]. Wang et al. identified miRNAs whose expression is specific to men of African ancestry, and the genes they regulate include miR-133a/*MCL1,* miR-513c/*STAT1*, miR-96/*FOXO3A*, miR-145/*ITPR2*, and miR-34a/*PPP2R2A*. The results of the study also suggested that the changes in these miRNAs activate EGFR–PI3K–AKT signaling pathways while knockdown of these miRNAs results in decreased proliferation, aggression and sensitivity to docetaxel-induced cytotoxicity [102].

A study examining the differences in miRNA profiles between healthy women and women with type II ECa, identified 280 miRNAs whose expression was different in the two groups. Women of African ancestry with ECa had increased expression of miR-1269b and decreased expression of miR-1269a, miR-891a and miR-892a compared to women of European ancestry [21]. The miRNA hsa-miR-337-3p was found to be downregulated more often in women of European ancestry with type II ECa [103]. The decrease in the levels of this miRNA is associated with lymph node metastasis in cancers such as gastric cancer [104].

### 5.2. PCa and Type II ECa Associated Long Non-Coding RNAs (lncRNAs) in the African Population

Long non-coding RNAs (lncRNAs), which are transcripts that are >200 nucleotides long and have no protein-coding potential, have emerged as important targets in tumorigenesis and tumor progression studies. Aberrations in the transcription profiles of various lncRNAs have been shown to be the driving force behind several cancer phenotypes [105]. They do this through their interactions with other components of the cell such as proteins, DNA and other RNA molecules. Increasing evidence shows that lncRNAs have great potential to be diagnostic and prognostic biomarkers because they are expressed in a tissue, cell type and cancer type-specific manner [106] (Figure 5). LncRNAs are detectable in bodily fluids and cancer samples; therefore, they have appreciable value as diagnostic tools and as potential biomarkers [107]. The most widely used test for the detection of prostate cancer, the prostate-specific antigen (PSA) test, can produce false positives or negatives as several other disorders (e.g., benign prostatic hyperplasia) can raise serum PSA levels [108]. Thus, more effective biomarkers are needed. Differential display 3 (*DD3*), also known as prostate cancer antigen 3 (*PCA3*), is an lncRNA that was shown to be significantly overexpressed in PCa [109]. It has since become an FDA approved biomarker that has higher specificity than PSA [110].

In endometrial cancer, the lncRNA H19 has been shown to be significantly overexpressed in 60% of EC [111]. Its expression levels are known to increase with the progression of tumor grade. In a study by Peng et al. [112], it was shown that upregulation of H19 is linked to poor prognosis in The Cancer Genome Atlas (TCGA) datasets. This suggests that H19 may be a suitable diagnostics biomarker and a potential therapeutic target for endometrial cancer [107].

The use of lncRNA as biomarkers and therapeutic targets for PCa and type II ECa in individuals with African ancestry has great potential in precision oncology [92,93].

## 6. Possible Role Players in Cancer Disparities: Insights into Improvements

Most clinical trial studies have been done in patients from high-income countries (HICs). Furthermore, as previously discussed, emerging evidence indicates that, in addition to the environmental factors, genetic ancestry and geographical factors also contribute to the diverse molecular landscape of disease [113]. In addition, there is existing evidence that those fundamental cellular processes differ across diverse populations. As the application of precision oncology is advancing in HICs, extensive efforts are being made to implement precision medicine on a global scale [114,115]. These efforts must be supported by all stakeholders including policy makers, funders and researchers. There are already existing barriers to improving the quality of health in many diseases, including prostate cancer and type II endometrial cancer patients in African populations [114,115]. These include advanced disease presentation, delayed healthcare access, limited access to treatment, cultural and religious barriers, education, and socioeconomic status. It has been reported that distinct human populations have distinct genetic variations that include germline, somatic and epigenetic alterations [114,115].

This suggests that there may be significant genetic differences in tumor types found in HICs and LMICs [116,117,118,119,120]. These population specific genetic disparities may be behind high PCa and type II ECa mortality rates in the African population. Furthermore, it has been reported that, in the USA, cancers in the African-American population show about 25% increased mortality rates compared to their white counterparts [121]. Additionally, oncogenes have also been shown to give rise to distinct mutations in different racial groups. For example, in ECa, p53 mutations are more common in women of African ancestry, while PTEN mutations are common in women of Caucasian or Asian ancestry. These oncogene mutations are reported to also occur at the nucleotide level [21]. LMICs such as the countries in the SSA region are not adequately represented in the International Cancer Genome Consortium (ICGC). Unfortunately, these disparities limit the applicability of findings emanating from such significant efforts [122].

Even though studies from HICs reveal major differences in toxicities and the response to treatment across various ethnic groups, genomic data from LMICs are still scarce. Differences in polymorphisms across different ethnic groups also play a role in precision oncology [122]. For example, African-Americans are predisposed to hematological toxicities associated with 5-Fluorouracil (5FU), and are more likely to have Thymidylate synthase (TYMS) and Dihydropyrimidine dehydrogenase (DPYD) gene variants, compared to their white counterparts who are more likely to suffer from 5FU associated nausea, vomiting, diarrhea and mucositis [122]. Similarly, it has also been reported that Doxorubicin metabolism by African Americans is severe, leading to cardiotoxicities, compared to their white counterparts. Furthermore, African-American polymorphisms may also be associated with acute toxicities such as neutropenia, as declining neutrophil counts are more commonly observed following chemotherapy in patients of African-American and Asian descent, compared to patients of European descent [123]. The Human Hereditary and Health in Africa (H3Africa) project is a collaboration between African clinicians, scientists and bioinformaticians, who perform large-scale genetic sequencing studies [124]. Even though data from HICs are deposited into this project, much data are still required from the LMICs. This highlights the significance of bottom-to-top approaches for precision medicine and oncology from the population level, particularly in the LMICs [125,126].

Despite efforts made by the National Institutes of Health (NIH) Revitalization Act in 1993, instructing the inclusion of underrepresented/minority populations and women in clinical trials, persistent exclusion or minimal inclusion of such minority groups/populations continue to arise [127].

The reluctance of individuals from specific population groups to participate in research and clinical trials may partially be attributed to historical unethical practices and fear of exploitation. Awareness campaigns, protection, privacy and transparency policies may also need to be reevaluated. Furthermore, studies illustrate that research in health disparities is inadequately funded [127,128,129].

Historically, African ancestry linked populations have been underrepresented in clinical trials. Adequate representation of Africans could result in new treatments that would benefit their overall quality of health and prolong their lifespan [44,45,46,47]. It has been proposed that the lack of research evidence on the genomics and application of personalised medicine in African groups could be solved through the study of African American populations. These individuals may carry similar genetic traits to Africans, but since they are in a developed country with better funded research projects, this has not proved to be the case. Historically, African Americans have been reluctant to take part in research projects; this is due to this population group having an attitude of fear and mistrust towards medical research. These fears and mistrust are not unfounded [121]. There are many examples of unethical experimentation and treatment of African Americans by the medical research community [130]. A prime example of this is the Tuskegee Syphilis Study initiated in the 1930s. Poor black sharecroppers from Macon County, Alabama with syphilis and matching controls were studied in order to determine the natural history of syphilis. When the study started, there was no standard effective treatment for syphilis. Penicillin became an effective treatment in the 1940s. However, the researchers did not treat the participants in the study and did not inform the patients that they were suffering from syphilis. This continued until the 1970s when the study was ended due to exposure by the mainstream media [131]. A study was conducted in order to determine why African Americans are underrepresented in medical studies. This study concluded that there were four main reasons why this occurred. These four major reasons were a general lack of awareness of trials, mistrust of the medical system, economic factors, and communication gaps [44].

## 7. Cancer Genomics Research in LMICs: Challenges and Opportunities

Concerted efforts involving clinical, basic and translational research have made significant progress in defeating cancer. However, cancer still remains a public health problem both in developed and LMICs [132]. Furthermore, the global health inequalities are exacerbated by large current oncology research, which is largely supported by the non-public/pharmaceutical industry. Commonly, such research mainly addresses industry-related questions and does not adequately address health problems in the underrepresented populations. LMICs represent most of the world population and, unfortunately, cancer incidence and mortality rates in LMICs are increasing significantly. Conversely, there is minimal to a complete lack of representation of LMICs, including African populations in ongoing oncology clinical trials [133]. Although the main goal of oncology clinical trials is to increase the real-world positive outcome for patients, the cost associated with this process is high. It is estimated that new drug research and development costs have been increasing over the past decades and current costs between $200 million to ~$3 billion [134,135]. These recent high costs of research and development of new medicines, associated with stringent regulatory processes, are contributing factors to excluding academic participation from clinical trial research, particularly in LMICs. In the implementation of precision medicine/oncology in the LMICs, cost is a significant hindering factor. Even though precision oncology costs may hold the potential to be cost-effective, it may be difficult to recoup these costs as they may possibly be higher than the traditional treatment [136,137]. However, the cost limiting factor can be overcome by the generation of biosimilar or generic drugs, as has been similarly applied with the HIV antiretroviral drugs [138]. In addition to the cost impeding factor, the sub-optimal functionality of the electronic medical record system (EMR) may be another limiting factor. The EMR system can be ideal for storing and referencing relevant patient pharmacogenomics’ data, as this will play a crucial role in suitable drug selection [4].

The genomic and molecular investigation of a variety of tumors has initiated the development of drugs targeting the discovered biomarkers. These molecular changes can either be driving the carcinogenic process (driver mutations) or be mere bystanders/passenger mutations [139,140]. Differentiating between these two forms of mutations can be complicated and requires development of molecular tumor boards (MTB), comprised of multidisciplinary experts enlisted to analyze the vast amounts of heterogeneous data generated through studying cancers [3]. Genomic-driven cancer management (precision oncology) is yet to be uniformly rolled out throughout the world, more especially in the so-called “Global South” where contributions to cancer genomic knowledge are lacking. This lack of genomic information from Africa complicates the implementation of precision oncology in this region with patients often relying on pharmaceutical company initiated clinical trials to access targeted therapies [3,141].

Precision oncology not only focuses on the cancer genome, but also on the transcriptome and proteome of the tumor. This improves the identification of novel biomarkers and therapeutic targets. However, if systemic bias is not properly addressed, such as the inclusion of the minority underrepresented populations, precision oncology may actually exacerbate the disparities that already exist in health care systems [142]. Through the collaborative worldwide genomic research efforts such as The Cancer Genome Atlas (TCGA), the International Cancer Genome Consortium (ICGC) and the Pan-Cancer Analysis of Whole Genomes (PCAWG), there has been significant progress to identify the genomic aberrations that underpin cancer biology [143,144,145].

## 8. Pan-African Genomics Cancer Research

The genetic differences between African and non-African populations include copy number variation, haplotype, and nucleotide diversity in mitochondrial and nuclear genomes [145,146,147]. The African populations harbor the most genetically diverse populations [147]. Furthermore, the SSA Khoisan ancestry has been linked to high risk PCa [148]. Thus, the inclusion of such populations in international cancer research and clinical trials offers even greater opportunities in the pursuit of precision oncology. Unfortunately, Africa has about a 4-fold increase in cancer mortality rates, compared to HICs [149,150]. The Human Heredity and Health in Africa (H3Africa) initiative involves projects such as Genome Wide Association Studies (GWAS) for breast and prostate cancers in the African American populations [151,152]. Unfortunately, the high risk loci identified in European genomes has not been linked to the PCa patients in Southern Africa, although European PCa genetic risk loci have also been identified in African Americans [153]. Current efforts to address the disparities in the data concerning the genetic basis of PCa include the establishment of the Men of African Descent Carcinoma of the prostate (MADCaP) Network in SSA [154]. This network aims to develop cancer genomics methods in the SSA region. Lack of efficient cancer registries in the SSA region also impedes functional cancer genomics research in this area. The establishment of the African Cancer Registry Network (AFRCN) in the SSA region in 2012 was to collate the already existing cancer registries in this region and promote the establishment of new ones [154]. Prioritization of international cancer research funds in LMICs, inclusion of racial, geographical, socio-economic, cultural and education would benefit the African populations. In this way, increased diversity in cancer research and clinical trials will pave the way for precision oncology to become the basis for an era of increased research and discovery in African populations.

## 9. Conclusions

There is a fundamental need for the inclusion of LMICs, such as those located in the SSA region, in global cancer genomic studies. Apart from benefiting cancer patients in this region, the SSA populations hold enriched genetic diversity which would be instrumental in the continued fight against cancer on a global scale. Even though PCa and type II ECa are linked to African ancestry, much is still unknown. This is why decoding of the genomes of members of this population is necessary to embrace precision oncology. The advantages and implementation of precision oncology are summarized in Figure 6.

The burden of PCa on men of African ancestry and ECa type II on women of African ancestry, the great genetic diversity in SSA, the inadequate efficiency of diagnosis-to-cure using conventional treatments for advanced PCa and type II ECa with their known side effects as well as the high costs of some of the more effective treatment strategies available, make the search for affordable, precise medicine an imperative for the African continent. In the pursuit of medical care with a strict adherence to the ethical principle of first do no harm, the clinicians and scientists’ oncology experts are compelled to continue seeking alternatives for the cure and the care of PCa and ECa type II patients in the African region, while being cognitive of the potential associated high costs. At this current point in time, the ability of precision oncology to ease the burden of PCa and ECa type II in the African population may encounter implementation challenges. However, the benefits of this approach will benefit patients of African ancestry in many ways that will justify the time and money spent to develop and implement precision oncology. This calls for the urgent implementation of mechanisms to include populations of African ancestry in international efforts involving genomics to combat heterogeneous diseases such as cancer.

## Figures and Tables

**Figure 1 ijms-23-00628-f001:**
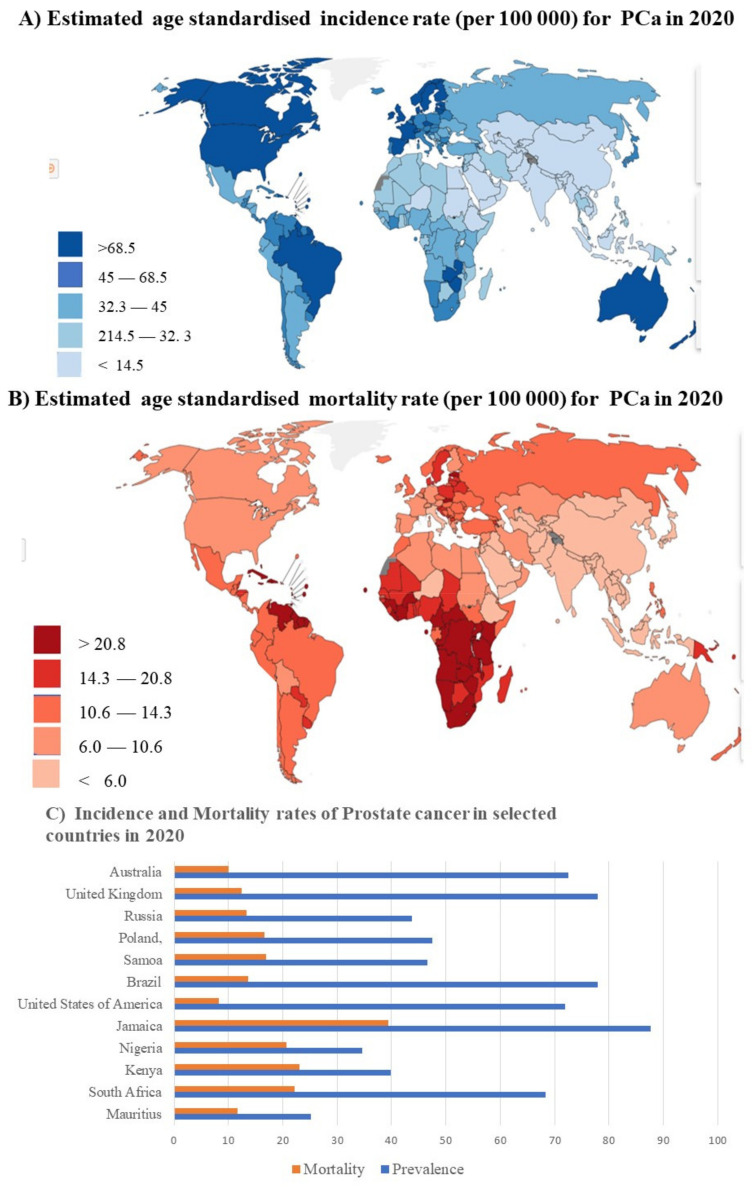
GLOBOCAN incidence and mortality statistics for Prostate cancer: (**A**) world map representing the worldwide age standardized incidence rates of prostate cancer. The highest rates are observed to be in North America, the Caribbean, Western Europe, Brazil, Australia and New Zealand; (**B**) world map representing the worldwide age standardized mortality rates of prostate cancer. The highest mortality rates are in Sub-Saharan Africa; (**C**) a graph comparing the incidence and mortality rates of specific countries demonstrating various trends. All data fall within the 95% confidence interval [28].

**Figure 2 ijms-23-00628-f002:**
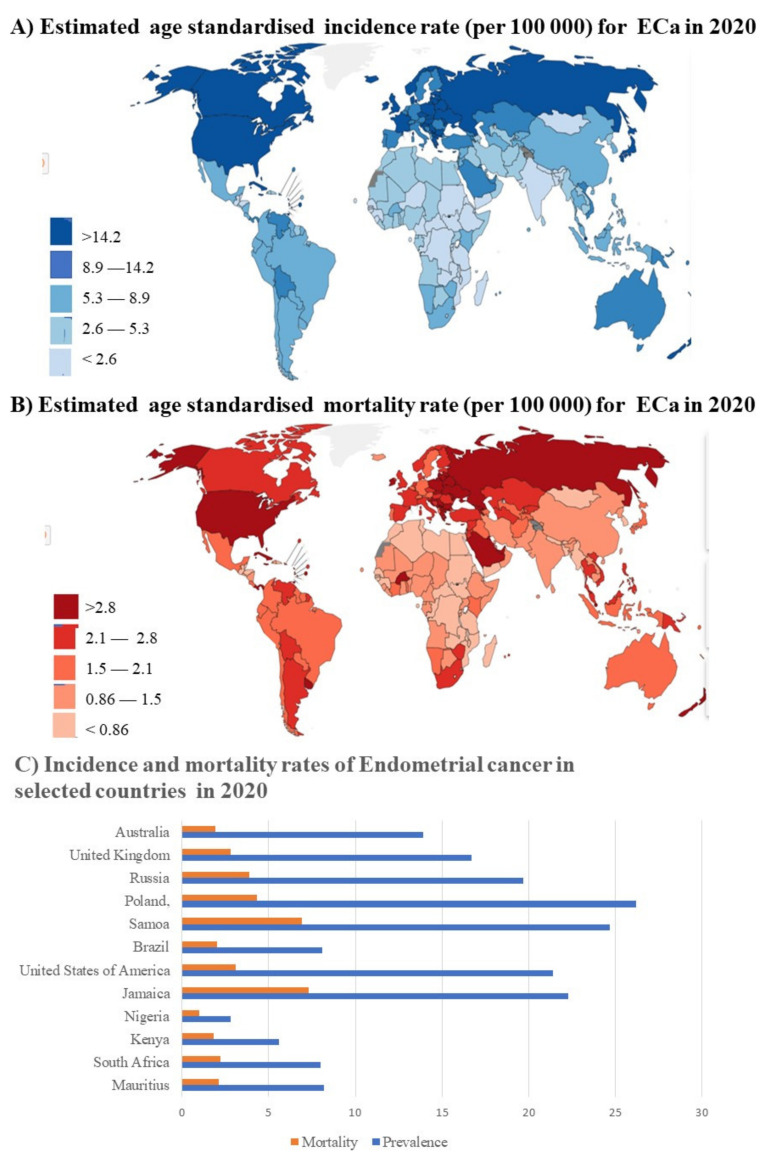
GLOBOCAN incidence and mortality statistics for Endometrial cancer: (**A**) world map representing the worldwide age standardized incidence rates of endometrial cancer. The highest rates are observed to be in North America, Eurasia, Australia and New Zealand; (**B**) world map representing the worldwide age standardized mortality rates of endometrial cancer. The highest mortality rates are in North America and Eurasia, mirroring the incidence rates and Australia and New Zealand; (**C**) a graph comparing the incidence and mortality rates of endometrial cancers in specific countries demonstrating various trends. All data fall within the 95% confidence interval [28].

**Figure 3 ijms-23-00628-f003:**
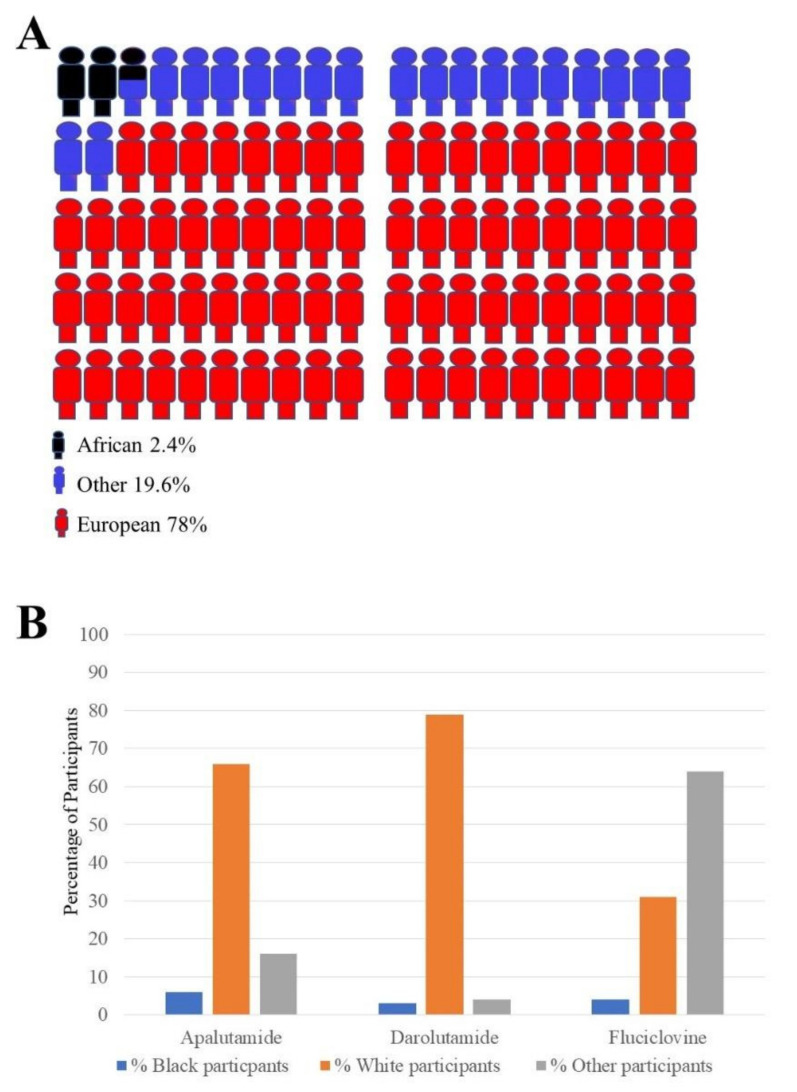
Racial disparitis in genome wide association studies (GWAS) and clinical trials enrolment: (**A**) Different populations’ contributions to new genomic related discoveries. Compared to other populations such the European group or other population groups, there is little participation of the African population in genomic related studies. These numbers come from participants taking part in a large number of studies (*n* = 110291); (**B**) summary of enrolment of black and white men in representative clinical trials for three Food and Drug Administration (FDA)-approved PCa drugs. The number of white participants enrolled for all three PCa FDA approved drugs (2016–2019) appears to be higher than the number of black participants in all three studies. (Apalutamide study *n* = 1207) (Darolutamide study *n* = 1509) (Fluciclovine study *n* = 596).

**Figure 4 ijms-23-00628-f004:**
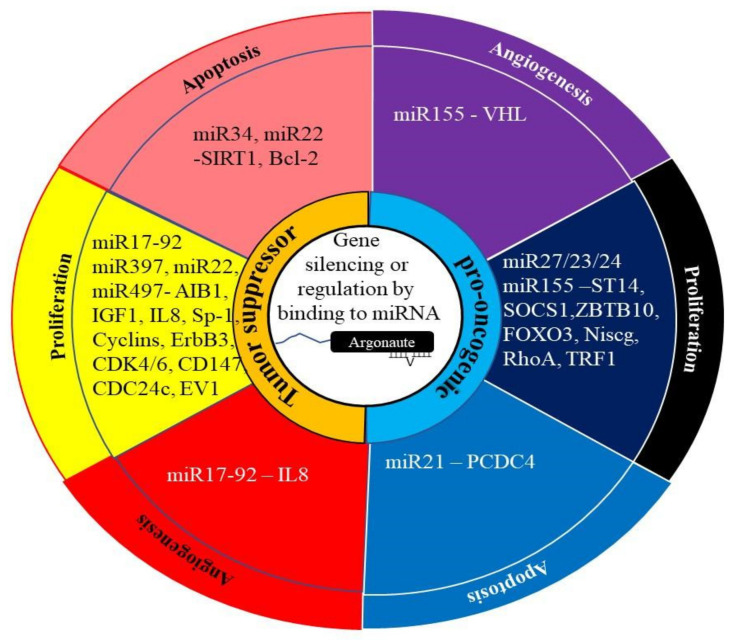
The role of miRNA in cancer. The regulation of gene expression by miRNAs generally involves them binding to their target mRNA and preventing or altering its translation into protein. This can be achieved through the degradation of the mRNA or altering splicing of the mRNA. In cancer, these miRNAs can act as pro-oncogenic by targeting tumor suppressor genes. Alternately, they can act as tumor suppressor miRNAs by targeting the miRNA of oncogenes.

**Figure 5 ijms-23-00628-f005:**
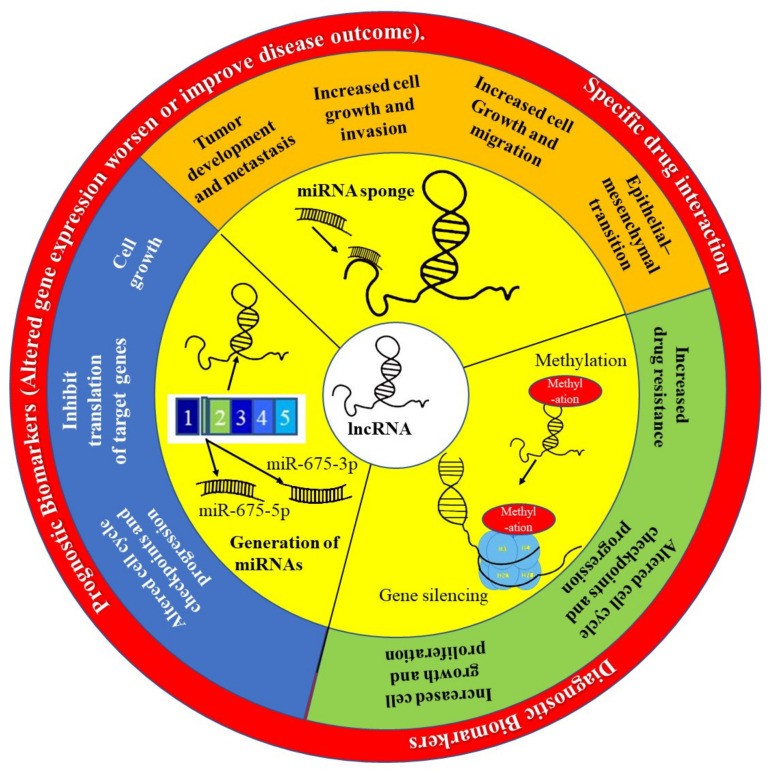
The role of lncRNA in precision oncology. The lncRNA H19 has been implicated as playing an important role in the development and progression of many cancers, including PCa. This lncRNA is able to act by preventing many miRNAs from performing their function by acting like a sponge and binding to these miRNAs preventing them from targeting MRNAs. The H19 gene is also spliced to generate miRNAs such as miR-675-3p and miR-675-5p. Finally, H19 can silence gene expression through methylation of histones.

**Figure 6 ijms-23-00628-f006:**
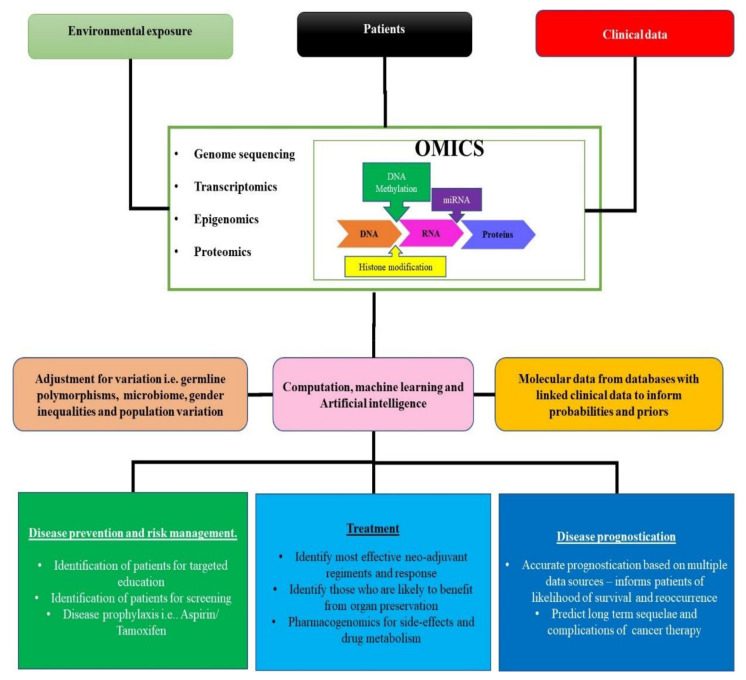
Precision medicine enabling technologies. New technologies to characterize the profiles of genetic mutations, transcriptome changes, epigenome changes and proteomic changes that are specific to various populations and diseases such as PCa or ECa have been developed. These techniques give rise to vast amounts of data. In order to analyse and curate these data, there are new machine learning and artificial intelligence networks and algorithms available. This will also allow the integration of this vast amount of sequencing data with data from other sources. The end results of the generation and analysis of this data will be the ability to implement precision medicine, resulting in improved disease prevention, management, treatment, and prognostic predictions.

**Table 1 ijms-23-00628-t001:** Prostate cancer population risk associated loci.

Gene Marker	Gene Product Role	Loci	Ref.
European only			
*CABP*	Calcium-binding protein 1	1p36	[51]
*HOXB13* rs138213197	Homeobox protein Hox-B13	17q21	[52]
European and African American			
*HPC20*	hereditary prostate cancer genetic-susceptibility locus	*20q13	[53,54]
*HPC1*	hereditary prostate cancer genetic-susceptibility locus	*1q24-25	[53,54,55,56]
*PCAP*	Predisposing for Cancer Prostate locus	*1q42-43	[53,54,56]
*HPCX*	Hereditary Prostate Cancer, X-Linked	*Xq27-28	[54,57]
		*8q24	[43]
		*3p12	[43]
*KLK2/3*	Kallikrein-2/3	*19q13.33	[42]
*NUDT10/11*	Nucleoside diphosphate-linked moiety X motif 10/11 (Nudix motif 10/11)	*Xp11.22	[49]
		11q13.2	[49]
*HNF1B/TCF2*	Hepatocyte nuclear factor 1-beta/Transcription gactor 2	17q12	[49]
*JAZF1*	Juxtaposed with another zinc finger protein 1		[50]
*MSMB*	Beta-microseminoprotein		[50]
African American			
*DXS986*	DExD/H-Box Helicase 58	*Xq21	[58]
*D17S1852*	Microsatellite marker	*17p11	[58]
*rs980481*	A/C/T single-nucleotide variation on chromosome 2	*2p16	[59]
*rs71527*	C/T single nucleotide variation affecting the gene coding for Carbamoyl-phosphate synthase 1 (CPS1)	*2p16	[59]
*rs11067228*	A/G single-nucleotide variation on chromosome 12	*12q24	[59]
*D11S908*	DNA segment containing a CA repeat	*11q22	[58]
*D2S2259*	DNA segment containing a CA repeat	*2p21	[58]

* PCa loci linked to African ancestry.

## Data Availability

Not applicable.

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
