# Peer review of "Role of Precision Oncology in Type II Endometrial and Prostate Cancers in the African Population: Global Cancer Genomics Disparities"

_ijms, 2022, doi:10.3390/ijms23020628_

Round 1

Reviewer 1 Report

The present work offers a general overview of precision oncology in type II endometrial and prostate 2 cancers. The review reads smoothly and clear and the authors highlight the main advantages of that kind of nanomaterials, including a high number of recent references. I list some minor concerns that need to be addressed.

  1. The description of the figures should be detailed in the manuscript, for example, Figure 1B need to give an explanation in the manuscript, there is only discussion with Figures 1A and C. The authors need to point to the paragraph of figure 3.
  2. The figures should incorporate statistical concepts and calculate the confidence interval to convince the phenomenon that there is a significant difference. For instance, different groups of participants need to make statistical
  3. The pattern mark in the center of Figure 4 is too small and unclear, and the quality of the pattern needs to be corrected.
  4. “These were* a general lack of awareness of trials, *mistrust of the medical system, * economic factors, and * communication gaps.” The meaning of the star mark in this sentence is unclear.

Reviewer 2 Report

In the submitted manuscript Marima et al. tried to presented an overview of precision oncology in type II endometrial and prostate cancers in the African population.

This manuscript has two main drawbacks: 1) it is unclear why only "type II endometrial and prostate cancers" have been reviewed, and 2) lots of statements are missing citations, therefore I cannot consider this manuscript as an original text.

Reviewer 3 Report

The current manuscript is a timely review of the racial disparity associated with prostate and endometrial cancer. This review also highlights the path forward to address such racial disparity and is acceptable in its present from for publication in the International Journal of Molecular Sciences.

Round 2

Reviewer 2 Report

The authors have improved their manuscript, however, these are still few things that must be corrected, clarified, uniformized and further improved.

1) Authors throughout the text often use terms LMICs and HICs but they did not explain what they actually represents!

In 'Introduction' they should briefly explain what those therms mean, how and by whom (World Bank) are countries classified, and (MOST IMPORTANT) they should properly used classification because, for instance, Russia is not any more considered as HIC (https://datatopics.worldbank.org/world-development-indicators/the-world-by-income-and-region.html)!

2) In 'Abstract', it is unclear what "redundant pathways" mean in the sentence "Particular attention is focused on activated/ repressed genes and associated pathways, while the redundant pathways are often ignored.".

3) On page 2, it is unclear what "target" means in the sentence "For example genes that increase a patient's risk of developing a specific target can...".

4) At the end of page 2 you wrote "Once again there is an absence of genomic data from Africans...", while on the next page you wrote "Despite the wealth of genetic data in the African populations...", so now it is not clear what is true?!

5) On page 6, it is unclear why references [165-168] succeed reference [37].

6) On page 6, there is no reference after very precise statement "Therapeutic clinical trials for men with PCa have considerably increased over the past 16 years.".

7) Figure 3 should be placed on page 7 after the paragraph which ends with text "...considered for adequate inclusion [39].".

8) On page 10, the proper name for specialized sampling tool is not "Pipeline" but "Pipelle"!

9) Uniformly use just a single acronym for the United States of America, either U.S.A. or USA, not US (e.g., page 10, second paragraph).

10) On page 11, it is not clear what did you mean by the phrase "As novel parts of the genome". I suppose you have meant by recently discovered part of (human) genome/transcriptome!

11) Please inspect miRBase (https://www.mirbase.org/) and learn how to properly write names of genes, stem-loop transcripts and mature microRNAs. For instance, miRNA26, miR1269b, miR1269a, miR891a, miR892a or hsa-mir-337-3p are all wrong!

12) Rearrange section "5.2. PCa and type II ECa associated Long non-coding RNAs (lncRNAs) in African population" in a way that you first mention lncRNAs related to PCa (PCA3) and then ECa (H19). Also, it is unclear why did you in Figure 5 legend mention H19 in the context of PCa and not ECa, while in the text you mentioned H19 in the context of ECa?!

13) On page 13, rephrase sentence "It has been reported that distinct human populations have genetic variations that include germline, somatic and epigenetic alterations" so that point would be on "distinct genetic variations", because EVERY human population has that types of genetic variations!

14) Recheck your manuscript that all used acronyms and abbreviations, even well known one like PET and CT, are explained.

Author Response

Response to Reviewer 2 comments for manuscript ijms-1502306:

Role of precision oncology in type II endometrial and prostate cancers in the African population: global cancer genomics dis-parities.

We would like to thank the reviewer for taking the time and effort to review our paper for a second time. We would also like to thank them for the useful and constructive comments

  • Comment:

Authors throughout the text often use terms LMICs and HICs but they did not explain what they actually represent!

In 'Introduction' they should briefly explain what those therms mean, how and by whom (World Bank) are countries classified, and (MOST IMPORTANT) they should properly use classification because, for instance, Russia is not any more considered as HIC (https://datatopics.worldbank.org/world-development-indicators/the-world-by-income-and-region.html)!

Response

Definitions of the terms have been given in the introduction. Countries mentioned throughout the paper have been classified based on the world bank website

  • Comment:

Comment: In 'Abstract', it is unclear what "redundant pathways" mean in the sentence "Particular attention is focused on activated/ repressed genes and associated pathways, while the redundant pathways are often ignored.".

Response:

This statement has been expanded to explain what is meant by redundant pathways.

  • Comment:

On page 2, it is unclear what "target" means in the sentence "For example genes that increase a patient's risk of developing a specific target can...".

Response:

This error has been corrected and the sentence has been restructured to correctly convey the intended meaning “For example specific target genes that increase a patient's risk of developing a certain cancer can assist in early diagnosis’’

  • Comment:

At the end of page 2 you wrote "Once again there is an absence of genomic data from Africans...", while on the next page you wrote "Despite the wealth of genetic data in the African populations...", so nCommentow it is not clear what is true?!

Response:

The second statement has been re-written in order to more clearly explain the fact the we are referring to the potential wealth of genetic information present in African genomes due to the great variation of these genomes “Despite the potential wealth of genetic data present in African populations, due to the high genetic diversity of African populations,  and LMICs with African populations, such groups/ populations are still underrepresented in the international platforms of genome-associated research studies [24].”

  • Comment:

On page 6, it is unclear why references [165-168] succeed reference [37].

Response:

The correct references have now been inserted

  • Comment:

On page 6, there is no reference after very precise statement "Therapeutic clinical trials for men with PCa have considerably increased over the past 16 years.".

Response

A reference for this statement has now been inserted

  • Comment

Figure 3 should be placed on page 7 after the paragraph which ends with text "...considered for adequate inclusion [39].".

Response:

The figure has been moved

8) Comment:

On page 10, the proper name for specialized sampling tool is not "Pipeline" but "Pipelle"!

Response:

The correct name has now been used

9) Comment:

Uniformly use just a single acronym for the United States of America, either U.S.A. or USA, not US (e.g., page 10, second paragraph).

Response:

The acronym U.S.A. has now been used uniformly throughout the paper

10) Comment:

On page 11, it is not clear what did you mean by the phrase "As novel parts of the genome". I suppose you have meant by recently discovered part of (human) genome/transcriptome!

Response:

This phrase has been re-worded to more clearly reflect our intended meaning “As recently discovered functional parts of the human genome, the non-coding RNAs”

11) Comment:

Please inspect miRBase (https://www.mirbase.org/) and learn how to properly write names of genes, stem-loop transcripts and mature microRNAs. For instance, miRNA26, miR1269b, miR1269a, miR891a, miR892a or hsa-mir-337-3p are all wrong!

Response

The names of the miRNAs have all been corrected as per the nomenclature rules

12) Comment:

Rearrange section "5.2. PCa and type II ECa associated Long non-coding RNAs (lncRNAs) in African population" in a way that you first mention lncRNAs related to PCa (PCA3) and then ECa (H19). Also, it is unclear why did you in Figure 5 legend mention H19 in the context of PCa and not ECa, while in the text you mentioned H19 in the context of ECa?!

Response:

This section has been rearranged. The reference to H19 in the centre has been removed as this was incorrect.

13) Comment

On page 13, rephrase sentence "It has been reported that distinct human populations have genetic variations that include germline, somatic and epigenetic alterations" so that point would be on "distinct genetic variations", because EVERY human population has those types of genetic variations!

Response:

The word distinct has been added to the statement to clarify that we are referring to distinct genotypes in distinct populations

14) Comment

Recheck your manuscript that all used acronyms and abbreviations, even well-known one like PET and CT, are explained.

Response

Acronyms have been defined the first time they are used throughout the paper

Round 3

Reviewer 2 Report

Authors have satisfactorily responded to all my questions and accordingly corrected their manuscript.